# Rehabilitation after Allogeneic Haematopoietic Stem Cell Transplantation: A Special Challenge

**DOI:** 10.3390/cancers13246187

**Published:** 2021-12-08

**Authors:** Hartmut Bertz

**Affiliations:** Department of Medicine I, Hematology/Oncology/Stem Cell Transplantation, Faculty of Medicine, Medical Center, University of Freiburg, D-79106 Freiburg, Germany; hartmut.bertz@uniklinik-freiburg.de

**Keywords:** alloHCT, rehabilitation, immunosuppression, graft-versus-host disease, exercise

## Abstract

**Simple Summary:**

After undergoing an allogeneic haematopoietic stem cell transplantation (alloHCT), patients need intensive physiological and psychological rehabilitation. This should start immediately after discharge from the transplant ward as in- or outpatient rehabilitation. The rehabilitation centres should be qualified and experienced because this patient group exhibits problems that differ from those of patients who have undergone oncological therapies or autologous HCT. An experienced multidisciplinary team in close consultation with the primary transplantation centre should perform the rehabilitation therapy. This review will show the special challenges of these patients according to different timepoints after HCT. Because there is so little data available, personal experience and general guidelines on patient care after alloHCT are presented.

**Abstract:**

The general population is getting older and suffer more haematological malignancies despite being physically fit. These malignancies are mainly only curable via an alloHCT, and they are now carried out more frequently. Patients benefit from intensive rehabilitation earlier and may need it repeatedly in cases of severe side effects (e.g., graft-versus-host disease). They can suffer many problems that other cancer patients do not experience, such as severe infections, continued immunosuppression, nutritional restrictions, acute or chronic GvHD, or organ impairments (e.g., lung, eyes). They may also encounter various associated psychological problems, e.g., feeling like a chimera. Rehabilitation centres willing to care for patients after alloHCT should have an experienced multidisciplinary team and should work in close co-operation with the primary transplant centre.

## 1. Introduction

In- and outpatient rehabilitation after completing cancer therapy is an established procedure with different guidelines in certain countries [1,2,3].

Cancer therapies include chemotherapy, radiation therapy or surgery, and more recently, different immunotherapies (checkpoint inhibition, CAR-T-cells) or so-called targeted therapies. The latter include new drugs that specifically target the development of cancer cells and suppress their proliferation. The advantages of such targeted therapies are that the side effects on the cancer patients’ bodies are significant weaker than those caused by the more toxic chemotherapy.

One special immunotherapy is the allogeneic haematopoietic stem cell transplantation (alloHCT) to cure patients diagnosed with aggressive leukemias or lymphomas. After the alloHCT procedure, patients can suffer severe side effects from the immunotherapy (e.g., so-called graft-versus-host Disease (GvHD)) and the continued use of necessary medication [4].

The rehabilitation of patients after alloHCT differs from “standard” rehabilitation, and it demands a lot of specialised experience on the part of physicians and nursing staff; physiotherapists, dietitians, and psycho-oncologists may also be confronted with extraordinary situations [5].

To carry out an alloHCT, an allogeneic transplant centre requires a European Accreditation (www.JACIE.org, accessed on 5 October 2021) from the European Society for Blood and Marrow Transplantation [6]. In Germany, they must perform at least 25 alloHCTs per year to be reimbursed by health insurance providers.

Therefore, the rehabilitation of a patient after alloHCT should likewise only be conducted in an experienced rehabilitation centre/clinic, ideally with a secondary or tertiary acute hospital close-by to transfer the patient to in case of emergencies.

As so little is known about the guidelines, needs, or recommendations around rehabilitating an alloHCT patient [2,7,8,9], this paper reviews the status quo, addresses the risks and characteristics of a typical patient after alloHCT, and illustrates this patient group’s needs for rehabilitation.

## 2. AlloHCT

In 2019, patients in Germany (*n* = 3395 (www.drst.de, accessed on 15 September 2021)) and the US (*n* = 9498 (www.cibmtr.org, accessed on 15 September 2021)), and in 2018, patients in Europe (*n* = 19.630 patients (www.ebmt.org, accessed on 21 September 2021)), underwent an alloHCT. The two main indications for an alloHCT are malignant haematopoietic diseases or immunodeficient diseases (Table 1).

The ages of these patients diagnosed with a disease in this list of alloHCT indications range from newborn to 80 years; this review, however, will deal only with adult patients (>17 years) and their potential for rehabilitation. Due to the transplant procedure’s side effects and associated high non-relapse mortality (NRM), only younger adults up to the age of 45 years were transplanted until 1998. After 1998, so-called reduced intensity conditioning preceding the transplantation was introduced, and the age of eligible patients rose without increasing the NRM [10,11]. The incidence of malignant diseases rises in the sixth or seventh decade of life, but these patients are now biologically fitter than 40 years ago. Therefore, more patients aged above 60 years who have various comorbidities—as well as side effects and implications for rehabilitation measures—are being transplanted [12,13]. The role of rehabilitation in this age group is no longer to enable the patient to return to work [14,15]; another caveat is that older people are more likely to have lost their partner, to be living alone, and to be not as socially integrated as patients that are working [13,16,17].

The alloHCT process entails determining whether the patient fulfils the indication for an alloHCT; it is fulfilled when the disease is not otherwise curable, and a suitable donor must be found. The donor may be an HLA-suitable (human leukocyte antigen)/identical family member, or a voluntary donor from one of the worldwide registries [18]. In the days before the transplantation, the patient undergoes an age- and disease-adapted chemo-/immuno-/radiation therapy, and on day zero, the graft from the donor containing around 5 × 10^6^ haematopoietic stem cells/kg/body weight. Over the following three weeks, this graft establishes both a new immune system (with B- and T-cells) and a new haematopoietic system (with leucocyte, platelets, and erythrocytes) in the patient’s bone marrow.

Soon after the transplantation, to prevent the most severe and lethal complications, namely the aforementioned GvHD, patients are given immunosuppressive drugs before and after the HCT. These in turn make the patient extremely vulnerable to infections during rehabilitation. How long this vulnerability persists depends on the length of time after the alloHCT and when rehabilitation starts.

## 3. Acute Rehabilitation as an Inpatient

A few transplantation centres transfer their alloHCT patients soon after haematological regeneration (stable engraftment of leucocytes >1000/µL and platelets >20,000/µL) to an inpatient rehabilitation clinic. These patients must be able to care for themselves, i.e., take their meals, and exercise, for example.

This transfer usually occurs around day +25 (ranging between 19 and 35 days) after the transplantation. These “early” patients suffer from the same side effects and discomforts as other cancer patients soon after therapy (Table 2), such as fatigue [19], nausea, vomiting, neurocognitive deficits [20,21], and perhaps diarrhoea. Additional specific problems after alloHCT are discussed below.

### 3.1. Malnutrition

These patients often have no appetite, and report being satiated/”full” soon after eating very little food. This is a side effect of continued immunosuppression: calcineurin-inhibitors in particular (CNI; cyclosporine, tacrolimus) lead to a kind of gastroparesis. No appetite and early satiation exacerbate the malnutrition patients’ experiences at day +30 after alloHCT [22], and some have to be readmitted [23]. Despite early acute nutrition counselling on the transplant ward, we demonstrated that patients lose weight, especially muscle mass, between their admission for alloHCT and discharge. At day +100 most showed little nutrition recovery. Another side effect of CNI is the loss of magnesium in the kidneys—without magnesium substitution, the patient experiences muscle cramps or seizures. However, oral substitution in high doses causes diarrhoea, which is again associated with malabsorption and fluid loss, worsening malnutrition at this time.

Later in the time course after alloHCT, the target level of the CNI serum doses are reduced, and less magnesium is necessary. To treat malnutrition, or rather to stabilise the patient’s weight during rehabilitation, intensive nutrition counselling is necessary through providing high-protein foods served in small portions [24,25]. Nevertheless, patients lack lactase soon after alloHCT (an enzyme produced in the gut’s epithelial cells) due to expired mucositis after the conditioning therapies. Therefore, as they tolerate fewer milk products, they should be given lactose-free products. Furthermore, any foods containing bacteria or moulds (like blue cheeses) should also be avoided. These patients have to follow special hygiene rules because of their continuous strong immunosuppression, as that puts them at a high risk for food-borne infections (Table 3). The kitchen at the rehabilitation institution should be able to provide special alloHCT meals. Special nutrition education sessions should be offered during the patients’ stay.

### 3.2. Muscle Loss

After admission to alloHCT, the patients stay about ~40 days on the transplantation ward, mainly in their single isolation room, usually lying in their bed; their physical activities diminish dramatically (to 10–15%) [26] and their muscles shrink. Additional drugs like corticosteroids and CNIs cause myopathy, which is then aggravated by the polyneuropathy induced by several drugs (e.g., CNIs). Paradigmatic change has happened over the last 25 years, and after the first evidence of its benefits was published [27], exercise was introduced on the transplantation wards, and the patients are now motivated to exercise. However, patients lose a lot of muscle tissue despite physiotherapy and special training in their rooms [28]. Developing muscles is a further challenge for the rehabilitation centre; patients feel weak, experience fatigue, are too exhausted to climb a few stairs, and have, due to the continued medication, worsening polyneuropathy. Three different kinds of exercise should be encouraged for patients:Endurance;Strength;Balance.

Training can be done in groups, but many patients after alloHCT need an individualised training program [29]. Whole body vibration has been introduced without major side effects on the transplantation ward; it increases the muscle tissue and improves functional capacity. It is also safe and effective [28,30] in the rehabilitation setting as well, as are Nordic walking, ergometric training, electro muscle stimulation (EMS) [31], and low-weight training. The main task during rehabilitation is to improve the patient’s physical performance so that they can go home and live without assistance in their own rooms. Special attention should be paid to the climbing of stairs, which is impaired by the aforementioned myopathy. One often neglected aspect is balance [32,33]; patients with balance problems [29,30] carry a high risk of falling, and osteoporosis leads to fractures. As infections should be avoided, training the breathing muscles is a further target of special exercise in this patient group [34] (Table 4). Because these patients are severely immunosuppressed, training in a group may be impossible because of the high risk of infections, which is why individualised training programs are preferable and should be offered.

### 3.3. Risk of Infections

Especially soon after transplantation, immunosuppressive strategies around and after the transplantation suppress patients’ immune system. The most impaired and vulnerable are patients who have undergone a haplo-identical or cord blood transplantation or graft from a non-family donor. Granulocytes may be in the normal range, but B- and T-lymphocytes are reduced, as are monocytes. Furthermore, polyvalent immunoglobulins have to be substituted to enhance the infection defence. A serum IgG level of >500 mg/dL should be targeted. To avoid infections or viral reactivation, patients have to take many different drugs, which must be continued until the end of immunosuppression; moreover, during rehabilitation, weekly laboratory diagnostics to detect CMV- or EBV-reactivation should be performed. The main infectious pathogens that play a key role during the early inpatient rehabilitation phase are listed with their prophylaxes in Table 5.

To reiterate—GvHD is a very serious complication after alloHCT. Every infection can trigger this immunoreaction [36]; therefore, preventing infections is one of our main challenges. Special hygiene (hand, body, and food) procedures should be implemented during rehabilitation, and the patient should be given specific recommendations for when they can return home. The upper respiratory tract (paranasal sinus, lung) is particularly vulnerable as an entrance pathway for pathogens (viruses, bacteria, and fungi), and AlloHCT patients should avoid direct contact with infected persons, group meetings, and exercise groups, especially during the viral infectious season (winter; influenza). They have no protection at all against these pathogens, and vaccination is started as early as 3–6 months after alloHCT (Table 6). In case of infection (clinical, fever, or C-reactive protein rise) antibiotic therapy should be initiated after sampling (blood cultures, urine, and swabs). An infection so soon after HCT can be dangerous, even lethal, because of the impaired efficacy of the splenic function against the encapsulated pathogens (streptococcus pneumoniae, haemophilus influenza, and meningococcus spp.)—pneumococcus in particular should be within the efficacy spectrum of the antibiotics applied [35]. In this early phase, many patients in rehabilitation have a device (port, central venous line) [37]; in case of signs of infection, a catheter-associated infection should always be suspected and addressed.

Early introduction of antibiotics reduces the risk of GvHD [36].

To avoid and/or reduce infectious complications in an alloHCT-patient, during their rehabilitation stay, all patients should also be advised to maintain strict hygiene rules at home, and be given written guidelines on food and beverage restrictions (Table 3). They should be prepared for vaccinations (Table 6) to start around month three, when immunosuppression is tapered.

### 3.4. Psycho-Oncological Aspects

In general, the psycho-oncological problems that patients face after alloHCT are the same as after other cancer therapies: depression, fatigue, body misfortune, loss of hair, less energy, fear, anxiety, and so on [38,39]; they also depend on the gender of the patient [40]. Nevertheless, there are a few different aspects to consider. As mentioned above, as patients still carry a high risk of infection after discharge from the transplant unit, they are afraid to engage in daily living activities (ADL). They do not know how to deal with their pets, or their flowers and plants at home, etc. Many questions about normal life arise. Of course, pets or plants are a source of infection, but they need not give them away; a high level of hygiene is important, and giving them specific guidelines can reduce their fear of infections. Furthermore, in the body of such patients is another human’s system, a new haematopoietic/immune system—the patient is now a so-called chimera. Some patients, or rather their relatives or spouse, find this situation difficult. This point should be addressed during psycho-oncological counselling. Personal experience shows that in the first 12 months after alloHCT, patients are looking forward to the future, to getting their social life back, to reducing their medication, and trying to recover. However, after one year, many suffer from more depression; they look back and see that they have survived a very dangerous procedure, and their mood changes. Psycho-oncological support becomes very important again.

Fatigue is the main problem these patients have and psychological support is the main therapy, but so too are acupuncture, massage therapy, and yoga/meditation. Note that all these immune-system-activating therapies (acupuncture, massage therapy, and complementary medicine) [41] should be carried out cautiously, to avoid eliciting or aggravating GvHD.

The move from the protected, secure transplant ward into the rehabilitation centre is a big step for some patients. They have spent about 40 days in intensive care and now they must care for themselves again; here is where they really need help and guidance. Moreover, they have been isolated for a very long period in a single room with few visits from relatives or spouses. This unusual situation can lead to marital discord and even divorce. This factor, as well sexual problems of course, should also be addressed during psycho-oncological counselling.

### 3.5. Psychosocial Aspects

One of the main goals of rehabilitation is to improve patients’ physical and emotional fitness for returning to work. The pension insurance companies mainly pay for the rehabilitation clinic stay. As mentioned, most alloHCT patients are older (median age of 57 years for all transplanted adults). That is because the incidence and prevalence of malignant haematological diseases are highest in the later decades of life. More of these are transplanted because their biological age in the 21st century is younger than 40 years before. For the psychosocial team, it is now their task to integrate patients back into social and family life. Many have had their malignant haematological disease for quite a long time (myeloproliferative neoplasms (MPN), myelodysplastic syndrome (MDS)) before undergoing alloHCT, or have been out of work for months because of the short timeframe of previous therapies (acute lymphoblastic leukaemia (ALL), acute myeloid leukaemia (AML)). These patients often have financial problems and need social assistance or welfare payments because of their reduced earning capacity, or they need guidance filling out a pension request, for example. This too should be addressed during rehabilitation [38,42]. Unlike some other rehabilitation patients, post-alloHCT individuals cannot immediately go back to work after the end of rehabilitation. One issue is their ongoing immunodeficiency, which makes them susceptible to common infections and toxicities at the workplace. These include chemicals that can trigger malignant diseases (e.g., benzole), and fumes or dusts, which can impair lung function. These psychosocial aspects should be kept in mind if, at the end of rehabilitation, terminating work is being discussed or recommended. Getting reintegrated at work by slowly increasing the number of working hours per week is an excellent tool to integrate younger patients in the workplace after alloHCT.

### 3.6. Graft-versus-Host Disease

GvHD is a particular feature of alloHCT [43]; it is the donor immune system’s B- and T-lymphocytes reacting against the host’s organ cells. The risk of suffering severe or mild GvHD depends largely on the patient’s human leucocyte antigen (HLA) identity [18].

We differentiate between two GvH reactions.

#### 3.6.1. Acute GvHD (aGvHD)

This can occur at the time of transplant engraftment when a few donor lymphocytes recognise that the host’s body tissue is not identical. It can occur until roughly a year after alloHCT but most arise between day +25 and +100 [44]. Its manifestation is an acute inflammation of mainly three organ systems: the skin, liver, and gastrointestinal tract. The skin looks severely sunburnt, laboratory anomalies are diagnosed, such as acute hepatitis or cholangitis, and the gut becomes inflamed such as in acute colitis, especially in the ileum (like Crohn’s disease), resulting in the loss of fluid and protein. Gastritis or duodenitis may occur as well. Gastrointestinal aGvHD can be an emergency depending on its extent and severity.

If the alloHCT patient is transferred early to a rehabilitation centre as discussed here, they should immediately inform the staff about skin changes or if their bowel movements are runny, too frequent, too much, or of a changing consistency. Both physicians and nurses should ask the patients on a daily basis about any changes and document them. Laboratory values should be closely tracked and monitored at regular time schedules.

If any signs of aGvHD occur, the immunosuppression should be adjusted, and if necessary increased, after contact with the transplant centre [45]. The patient may need to be readmitted to the transplant centre.

Later in the time course, once the immunosuppression has been tapered (starting around day +90–100) aGvHD can still occur as well.

#### 3.6.2. Chronic GvHD (cGvHD)

CGvHD can occur after day +100, develop out of aGvHD, or arise during immunosuppressive drug tapering [46]. Its manifestation is different; there is no inflammation, it is more of a “dry-syndrome”, as observed from different autoimmunological diseases [47]. Scleroderma is the skin manifestation, producing Sjögren-syndrome-like changes in the eyes and a dry mouth, or there can be fasciitis, myositis, obstructive lung disease, and so on. Nearly every immunological disease can be mimicked by cGvHD, and it affects nearly every organ system, except the heart. The problem of cGvHD for patients will be discussed later because they do not occur during early rehabilitation.

### 3.7. Laboratory Diagnostics

Frequent laboratory diagnostics should be performed during early rehabilitation. This includes the parameters listed in Table 7 at least twice a week; especially the blood levels of the immunosuppressive drugs. The transplant centre determines the appropriate drug levels of the immunosuppressives that they want for their patients (e.g., cyclosporine, tacrolimus, MMF/MPA, and Everolimus). In cases of divergent drug levels, the rehabilitation physicians have to adjust the drug doses. If the patient’s condition worsens, more frequent laboratory controls are necessary, especially in cases of fever or infection.

## 4. Acute Rehabilitation as an Outpatient

After spending about 40 traumatic days in isolation at hospital after undergoing an alloHCT, not every patient will want to be transferred to another clinic. Thus, there is the alternative of outpatient rehabilitation. Its requirements are:A compliant patient;The patient is mobile and can care for her/himself, or;Is well integrated and cared for by the family;Lives close to an alloHCT-experienced haematologist or;To the primary transplant centre’s outpatient department.

Outpatient rehabilitation in the first three months will primarily consist of physical exercise, as mentioned above, involving the training of strength, power, and balance [9,48]. This should be done at least twice a week at two- to three-day intervals. Again, the training should be managed on an individual performance basis. A few rehabilitation clinics offer training with their experienced team for outpatients as well but there are very few such clinics and not every local fitness centre is able to train these patients individually; moreover, they cannot ensure the essential hygiene conditions. An alternative is web--based training programs, which are individualised by the physiotherapists or sports scientists in the transplant centre. These programs can be adopted if the patient’s fitness is improving

Nutrition counselling in the transplant centre’s outpatient department on a regular schedule, as well as contact with a psycho-oncologist that is experienced in alloHCT, and the aforementioned special problems, is very important. Urgently recommended in cases of outpatient rehabilitation is that the patient also be examined twice a week by an experienced alloHCT physician, and that the laboratory controls listed in Table 7 take place. Furthermore, a 24 h telephone hotline is mandatory for this patient group. New in the last few years is a web or smartphone-based information system, whereby the patient can inform her/his caregivers every day about any clinical changes like diarrhoea, fever, vomiting, weight loss, and so on. The SMILE^®^ program [49,50] has been implemented in Freiburg with an excellent response, and high patient satisfaction. In cases of genuine emergencies, even at night-time, the patient must be able to be examined in an emergency department. Complications after alloHCT can be life-threatening within a few hours.

## 5. Acute Rehabilitation as an Inpatient Later in the Time Course

Because of the paucity of high-quality rehabilitation clinics specialising in, or qualified in, acute rehabilitation after alloHCT, many transplant centres discharge their patients into their own outpatient department. They try to provide outpatient rehabilitation programs, but they are often strapped financially. However, the guarantee of frequent and close controls, and follow-up exams of these patients in their centre during the first 100 days, is in fact more important than physical and psychological rehabilitation during this time period. The main reason for this difficult situation is simply the lack of rehabilitation facilities experienced in alloHCT treatment that are close to transplant centres [51].

If patients come in for rehabilitation later as an inpatient in their post-alloHCT time course (mainly on days +60–90), then their recovery has started, accompanied by the main side effects, and patients are more capable of participating in their tasks in the rehabilitation clinic—nutritional restrictions are less pronounced if no aGvHD is present. The laboratory controls may be reduced but the above-mentioned intensive training and care is absolutely essential.

The risk of an aGvHD persists and should be kept in mind during clinical examinations and lab controls, especially if immunosuppression is being tapered.

Because the immune system is slowly but steadily regenerating, the risk of severe infections is lower than in the first post-alloHCT months, but they still need to be diagnosed early.

These patients can participate in group exercise, lectures, and eat in the clinic restaurant. If they have not engaged in an intensive outpatient sports programme after discharge from the transplant centre, they will still be struggling with muscle loss, weakness, and a certain amount of fatigue. This should be dealt with intensively.

Generally speaking, alloHCT patients are usually fitter in this time period, but they can still suffer life-threatening medical emergencies within 24 h.

## 6. Rehabilitation with Chronic GvHD

The main challenges when caring for alloHCT patients in rehabilitation is the prevention of cGvHD. Its incidence can be minimised by leaving the patient’s immunosuppression medication on a high level, but the dilemma with that is that the risk of the relapse of their underlying malignancy then rises [4]. Long-term immunosuppression suppresses the graft-versus-leukaemia/lymphoma effect (GvL effect), which in turn increases the risk of a relapse. Around 40–60% of patients experience a cGvHD. How seriously they are affected depends on its severity (mild, moderate, and severe) and the organs involved (skin, fascia, joint, muscle, lung, and eyes). The diagnostic criteria of cGvHD, and its staging and therapy options, are actively and frequently discussed worldwide and constantly updated by the cGvHD consortium (www.gvhd.eu, accessed on 20 September 2021) [52]. It is important to keep detailed records and descriptions of the cGvHD before initiating or increasing immunosuppressive medication. These patients must be re-evaluated on a regular basis to assess the therapeutic success. In addition to the patient’s transplant, physician specialists in each involved organ should be frequently consulted for the patient (i.e., dermatologists, ophthalmologists, neurologists, and gynaecologists) [52,53,54,55].

What role does the rehabilitation centre play?

The main goal of a planned intensive rehabilitation period is physical therapy. The patients’ physical limitations are what mainly prevent them from participating in their ADL [56]. Such physical limitations affect the younger patient going to work, as well as older retired patients, and these impairments should be handled intensively [57]. To achieve this goal, the physiotherapist should possess a great deal of experience in treating this alloHCT complication in particular, especially in patients suffering from skin/fascia-related GvHDs [58]. A list of possible interventions is listed in Table 8. These should be done very often (not just once a week); furthermore, repeated intense weeks in the rehab-centre really helps these patients. This point needs to be discussed and justified with the insurance companies.

Such cGvHD-associated impairments can affect the patients emotionally, and psychologically as well; they cannot move as they used to (reduced performance), can suffer from shortness of breath [59] (which is extremely frightening), their appearance is altered (hair loss, dyspigmentation), and sexual activity is impaired in cases of cGvHD of the genitals (in females and males) [60]. These problems also require experienced psycho-oncologists because their treatment differs from the follow-up care of “normal” oncology patients.

With longer and more frequent support during a 3–4-week rehabilitation programme, these discomforts can be dealt with effectively. In our experience, in cases of severe cGvHD involving severe impairments, rehabilitation twice a year, or at the very least once a year, helps these patients.

Side effects of the CNIs are damage to the vessel endothelia, which leads to hypertension, and vascular diseases of the heart and brain. These impairments should be diagnosed and handled again mainly through exercise during follow-up care [61].

Depending on the dosage of immunosuppressive medication they have required, a patient can still carry a high risk of infections even years after alloHCT. The above-mentioned viral, fungal and bacterial prophylaxis medications should be continued and supervised. Long-term cGvHD patients carry an especially high risk of infections from moulds (especially aspergillus spp. and zygomycetes). Again, encapsuled bacteria pose an infection risk because of asplenia syndrome. The spleen is small, no longer functions well, and Howell-jolly-bodies appear in the blood smear. In case of infection, streptococcus pneumonia-active antibiotics should be initiated; frequent refresh-vaccinations are also strongly recommended.

Again, a multidisciplinary and highly experienced team is necessary to treat and care for cGvHD patients [62].

## 7. Conclusions

Rehabilitation—regardless of whether it occurs earlier or later—plays an essential role in the care of patients after they have undergone an alloHCT. Intensive rehabilitation should be performed during their inpatient stay. There are still no set guidelines on such rehab. In any case, therapy should be carried out by an experienced multidisciplinary team in close consultation with the primary transplantation centre. Patients are extremely grateful for such life-affirming help after the life-changing experience of an alloHCT with the prospect of a cure and a good quality of life [63].

## Figures and Tables

**Table 1 cancers-13-06187-t001:** Indications for alloHCT.

Disease
acute myeloblastic leukaemia myelodysplastic syndrome myeloproliferative neoplasia chronic myeloblastic leukaemia acute lymphoblastic leukaemia chronic lymphoblastic leukaemia Non-Hodgkin’s lymphoma Hodgkin’s disease multiple myeloma
severe aplastic anaemia inborn errors immune deficiencies autoimmune disorders sickle cell disease other non-malignant diseases

AlloHCT, allogeneic haematopoietic stem cell transplantation.

**Table 2 cancers-13-06187-t002:** Side effects of alloHCT.

Main Side Effects at Admission to Rehabilitation
fatigue nausea vomiting diarrhoea depression weight loss neurocognitive deficits

AlloHCT, allogeneic haematopoietic stem cell transplantation.

**Table 3 cancers-13-06187-t003:** Food to be avoided until day +180 after alloHCT.

Food to Avoid until Minimum Day 180 or Full Immune Reconstitution, Whichever Comes First
raw fish (sushi) raw milk and raw milk products raw meat (steak tartar, ground pork) carpaccio shellfish/crustaceans raw grain fresh or raw seeds or grains raw eggs mouldy food, blue cheeses soft ice grapefruit, pomelos, and pomegranates (they interfere with immunosuppressive drugs)

AlloHCT, allogeneic haematopoietic stem cell transplantation.

**Table 4 cancers-13-06187-t004:** Exercise to do after alloHCT.

Important Forms of Exercise
endurance training weight training balance training whole body vibration (WBV) electrical muscle stimulation (EMS) breathing exercises

AlloHCT, allogeneic haematopoietic stem cell transplantation.

**Table 5 cancers-13-06187-t005:** Infectious pathogens and their prophylaxes/diagnostics adopted from [35].

Pathogen	Drug Prophylaxis	Laboratory Diagnostic
CMV	Letermovir	CMV PCR weekly
EBV	No	EBV PCR weekly
HSV	Aciclovir/valaciclovir	HSV PCR if infect is suspected
VZV	Aciclovir/valaciclovir	VZV PCR if infect is suspected
Toxoplasmosis	Trimethoprim-sulfamethoxazole	no
Pneumocystis jirovecii	Trimethoprim-sulfamethoxazole.	no
Tuberculosis	INH + Vitamin B6	no
Hepatitis B	Lamivudine or entecavir	HBV PCR if reactivation is suspected

CMV cytomegalovirus, EBV Epstein Barr virus, HSV herpes simplex, VZV virus varicella zoster virus, INH isoniazid, PCR Polymerase Chain Reaction, HBV Hepatitis B virus.

**Table 6 cancers-13-06187-t006:** Vaccination Schedule adopted from [35].

Virus	Kind of Vaccination	Recommendation	Time Points after HCT	Antibody Titer before Vaccination
Tetanus	inactivated	recommended	month (6–)12	not recommended
Diphtheria	inactivated	recommended	month (6–)12	not recommended
Poliomyelitis	inactivated	recommended	month (6–)12	not recommended
Pneumococcus	inactivated	recommended	month (6–)12	not recommended
Pertussis	inactivated	recommended	month (6–)12	not recommended
Influenza	inactivated	recommended	>month 3, seasonal	not recommended
Haemophilus influenzae	inactivated	recommended	month (6–)12	not recommended
Meningococci	inactivated	recommended	month (6–)12	not recommended
SARS-CoV-19	inactivated	recommended	>month 3	unclear
Hepatitis A/B	inactivated	recommended	month (6–)12	not recommended
HPV	inactivated	recommended	month (6–)12	not recommended
Tick-borne	inactivated	recommended	month (6–)12	not recommended
Varicella	inactivated	recommended	month (6–)12	not recommended
Mumps	live vaccine	case-by-case decision	2 years; w/o IS	before vaccination
Measles	live vaccine	case-by-case decision	2 years; w/o IS	before vaccination
Rubella	live vaccine	case-by-case decision	2 years; w/o IS	before vaccination
Varicella	live vaccine	case-by-case decision	2 years; w/o IS	not recommended

HCT haematopoietic stem cell transplantation; HPV human papilloma virus; IS immunosuppression; SARS-CoV-19; w/o without.

**Table 7 cancers-13-06187-t007:** Laboratory controls during rehabilitation.

Lab-Controls Minimum Once a Week
white blood cell count, neutrophils, haemoglobin, platelets
potassium, sodium chloride, magnesium
ALT, gamma-GT
INR
C reactive protein
protein, serum albumin
immunosuppression blood levels

ALT, alanine aminotransferase; gamma-GT, gamma-glutamyl transferase.

**Table 8 cancers-13-06187-t008:** Physical therapy of chronic GvHD.

Special Exercises
massage breathing exercise connective tissue massage lymph drainage polyneuropathy training wraps light therapy with UVA A and B whole body vibration (WBV)

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
