# Peer review of "Rehabilitation after Allogeneic Haematopoietic Stem Cell Transplantation: A Special Challenge"

_cancers, 2021, doi:10.3390/cancers13246187_

Round 1

Reviewer 1 Report

The review article by Hartmut Bertz discussed in detail on some of the critical aspects of rehabilitation after allo-BMT and highlights the challenges in select patient population. The review article is well written and reads coherently. Minor edits;

  1. Line 60: patient range should be from newborn to 80
  2. Table 1: can include Sickle cell disease along with other NM disease, a specific group which would benefit with rehab
  3. table 2: title main side effect "at" admission to rehab
  4. table 3: food avoided until minimum day 180 or full immune reconstitution whichever is sooner
  5. Author should discuss about other novel rehabilitation approaches including massage therapy, acupuncture and yoga/meditation
  6. line 181: author should eliminate the line" early antibiotics reduces the risk of GVHD"- as this is contradicting with data supporting against use of prolonged antibiotics in early HCT period and association with gut dysbiosis and development of GVHD.

Author Response

Responses to Reviewer 1

Dear reviewer 1, thank you for your valuable comments and corrections:

@ Line 60: changed to newborn to 80

@ Table 1: sickle cell disease is added

@ Table 2: ad is changed to at

@ Table 3:  until minimum day 180 or full immune reconstitution whichever is sooner is added

@ Line 181: you are absolutely right with antibiotic therapy during the aplasia phase of alloHCT and dysbiosis as a trigger for acute, especially gut GvHD. But in the later time course of rehabilitation or in the outpatient department infections of the Sino-bronchial system should be treated early and brief with antibiotics. And Infections of devices should be treated as early as possible to avoid bacteremia

@ Native speaker: as queried by reviewer 2:

the paper is corrected for wording and grammatical quality by a native English speaker. The content and the messages of the manuscript are not changed

Reviewer 2 Report

Title:  Fine

Abstract: Content is fine

Main body of paper:

Section 3.2. Muscle Loss. What is the purpose of vibration training? It is noted to be safe, but what is the benefit of it (and evidence for the benefit, if any)?

Table 8. Special exercises. It would be good in the narrative content to learn about the reason for these modalities and any evidence for them.

References: fine

In general this manuscript contains excellent content for supportive care and safety strategies for stem cell transplant patients during acute rehabilitation, and can serve as an excellent resource.

In addition to the relative minor questions above, I really have just one major concern, and that is the grammatical quality of the writing. The paper does not appear to have been written by an individual for whom English is the primary language. The grammatical errors occur very frequently and are far too numerous to identify individually. While most are quite minor, for some phases, again too many to point out individually, it is difficult to be sure of the intended meaning. The paper should be thoroughly scrutinized by an expert individual with regard to accuracy of the English, in close collaboration with you to assure that your intended meaning is preserved. I hope you are able to do this because the content of this manuscript is highly informative and valuable.

Author Response

Responses to Reviewer 2

Dear reviewer 2, thank you for your valuable comments and corrections:

@ Whole body vibration: the concept of WBV and it evidence is shown in the cited literature [27,29]; ourselves performed a study showing the increase of muscles and the improvement of the functional capacity; it is an established method in many sports centers; I added increase of muscles and the improvement of the functional capacity in the sentence

@ Table 8: in guidelines for chronic GvHD; these therapy are recommended; there do not exist randomized trials to underline the effect of these therapies. I added the organs which may improve from these interventions according the special exercise

@ Native speaker: the paper is corrected for wording and grammatical quality by a native English speaker. The content and the messages of the manuscript are not changed

Round 2

Reviewer 1 Report

The authors have addressed all my queries. I have no further comments/suggestion. 

Author Response

I thank the reviewer for the primary helpful comments

Reviewer 2 Report

The paper is much improved! There are still a few relatively minor grammatical irregularities which can be addressed by editorial staff. My only other concern is that a few points would benefit from having references, including:

1) Line 121 ("physical activities diminish to 10-15%"; also, what does this mean, do you mean the patients' physical activities diminish to 10-15% of pre-transplant baseline?)

2) Lines 173-174 (about infection being a trigger for GvHD)

3) Line 190 (early anti biotics reducing GvHD risk)

4) Table 6 (Is this material from reference 34? not currently clearly stated)

5) Lines 217-219. What exactly do you mean by "immune system activating therapies"? Please give possible examples, and also the reference (or, if empiric, state that).

6) Lines 352-356 (greater risk of relapse with more prolonged immunosuppressant use).

7) Lines 386-387. 

Regarding references for the above points, it's possible that you can use your existing references for some of them. Also if you do not have references for making these points, please indicate that you are speaking empirically from your experience.

Also, line 242-- Please insert the word "some" other rehabilitation patients. In my experience, very few patients go immediately back to work at the end of rehabilitation, especially after inpatient rehabilitation, though I recognize that alloHCT patients could likely take longer than the typical patient to get back to work. 

Author Response

Dear, Reviewer 2 thank you for your valuable comments, especially and that the manuscript improved

Answers and my comments to your suggestions are:

@ Line 121 ("physical activities diminish to 10-15%"; also, what does this mean, do you mean the patients' physical activities diminish to 10-15% of pre-transplant baseline?):

Yes, we have our own data from a doctorals thesis, but they are not published

  • This sentence is added in the MS: not published own experience in a doctors thesis

@ Lines 173-174 (about infection being a trigger for GvHD)

  • A publication is added as reference number 35

@ Line 220 (early anti biotics reducing GvHD risk)

  • same publication is added as reference number 35

@ Table 6 (Is this material from reference 34? not currently clearly stated) Yes, it is.

  • Vaccination Schedule adopted from [34]

@ Lines 217-219. What exactly do you mean by "immune system activating therapies"? Please give possible examples, and also the reference (or, if empiric, state that).

  • Massage therapy, acupuncture, CAM are immune modulating and stimulating therapies; the sentence is expanded and a refrence (4) added.

these immune-system activating therapies (acupuncture, massage therapy, complementary medicine) [40]

@ Lines 352-356 (greater risk of relapse with more prolonged immunosuppressant use).

A reference regarding GvHD prophylaxis and risk of relapse is added number [4]

 @ Lines 386-387. There is no literature; it is personal experience

  • In our experience is a supplemented

@ Also, line 242-- Please insert the word "some" other rehabilitation patients. In my experience, very few patients go immediately back to work at the end of rehabilitation, especially after inpatient rehabilitation, though I recognize that alloHCT patients could likely take longer than the typical patient to get back to work. 

  • The word some is added

Thank you for your evaluation
